# Decoupling of Dual-Polarized Antenna Arrays Using Non-Resonant Metasurface

**DOI:** 10.3390/s23010152

**Published:** 2022-12-23

**Authors:** Shengyuan Luo, Peng Mei, Yiming Zhang, Gert Frølund Pedersen, Shuai Zhang

**Affiliations:** 1Antenna, Propagations and Millimeter-Wave Systems (APMS) Section, Aalborg University, 9220 Aalborg, Denmark; 2Guangdong Provincial Key Laboratory of Optoelectronic Information Processing Chips and Systems, School of Electronics and Information Technology, Sun Yat-sen University, Guangzhou 510006, China

**Keywords:** mutual decoupling, non-resonant metasurface (NRMS), wideband, dual-polarized, large-scale antenna arrays

## Abstract

A non-resonant metasurface (NRMS) concept is reported in this paper to improve the isolation of dual-polarized and wideband large-scale antenna arrays. By properly designing the NRMS, it can perform stable negative permeability and positive permittivity along the tangential direction of the NRMS within a wide band, which can be fully employed to suppress the mutual couplings of large-scale antenna arrays. At the same time, the proposed NRMS can also result in positive permittivity and permeability along the normal direction of the NRMS, which guarantees the free propagation of electromagnetic waves from antenna arrays along the normal direction. For demonstration, a 4×4 dual-polarized antenna array loading with the proposed NRMS is designed to improve the isolations of the antenna array. The simulations demonstrate that the isolations among all ports are over 24 dB from 4.36 to 4.94 GHz, which are experimentally verified by the measured results. Moreover, the radiation patterns of antenna elements are still maintained after leveraging the proposed NRMS. Due to the simple structure of the proposed NRMS, it is very promising to be widely employed for massive MIMO antenna arrays.

## 1. Introduction

Large-scale antenna arrays, consisting of numerous antenna elements, are promising candidates for the 5G and future 6G applications [1,2,3,4,5,6,7,8]. To maintain the performance of large-scale antenna arrays, mutual couplings among antenna elements should be much considered, as poor couplings will seriously deteriorate the antenna performance, such as power amplifier efficiency, active VSWR, channel capacity, radiation efficiency, impedance match, and signal-to-noise ratio [9,10,11,12,13,14,15].

In the past few decades, researchers have made much effort to reduce the interferences between array elements, and different methods have been proposed [16,17,18,19,20,21]. The first method is directly suppressing the propagation of surface wave coupling and space wave coupling, such as electromagnetic bandgap (EBG) structures [16], defected ground (DGS) [17], and resonators [18]. These methods can effectively reduce the coupling of the array based on the frequency responses of these decoupling structures, yet these decoupling structures usually work in a narrow band. These decoupling structures are also complicated and bulky to achieve the desired frequency responses for mutual coupling reductions and must be inserted between array elements, which need relatively large space. On the other hand, due to the frequency responses of the decoupling structures being polarization-dependent, they are not feasible for application in dual-polarized antenna arrays. As a result, the decoupling methods mentioned above are difficult for extension to massive MIMO arrays. The second solution is to introduce an extra coupling path to cancel the original coupling, such as neutralization lines [19] and parasitic element structure [20,21], which suffer from a strict requirement for the volume to achieve integration with antenna arrays. Therefore, it is hard to be used in large-scale arrays as well. At the same time, most of these decoupling structures are applied in single-polarized antenna arrays. Some of the literature on the decoupling of dual-polarized and large-scale antenna arrays has been reported [22,23,24,25,26]. The decoupling network [22,23,24] is an effective method. The decoupling network in [24] can reduce the coupling of the array from −15 dB to −28 dB. Notably, the decoupling network has a completely different working principle from the neutralization line, which introduces transmission lines and parallel reactance to enable the transmission admittance to be zero, thus achieving the purpose of decoupling. However, the decoupling network commonly needs a matching network to make the antenna achieve good impedance matching. It cannot be used in wideband arrays owing to the feature of resonant-based frequency responses of the decoupling network. A decoupling surface (ADS) is proposed in [25,26], which consists of a group of primary and accessorial reflectors to create partial reflective waves to cancel the unwanted coupling waves between array elements. however, constructing proper metal patterns of the ADS to cancel the reflecting waves has a complicated design process. In [26], a 4 × 4 staggered array with diversified phase laggings and coupling level lower than −25 dB is designed by loading the ADS. In [27], the decoupling dielectric stubs (DDS) are proposed to reduce the mutual couplings of a 4 × 4 antenna array below −25 dB from 4.4 to 4.8 GHz. However, it requires a relatively high profile (half wavelength). In [28], a decoupling ground (DG) method is presented. An improvement of about 7 dB isolation has been obtained with the inner-element distance around 0.62λ. Nevertheless, it needs a large inter-element distance to enable the space wave coupling and surface wave coupling with the same amplitude and opposite phase to perfectly cancel each other. 

Metasurface has been proposed very recently as a promising solution to reduce the mutual coupling of antenna arrays. In [29,30], a metasurface has been introduced to create a region with negative permeability and positive permittivity to suppress the coupling waves of two-element antenna arrays. A permittivity–negative metasurface superstrate was studied to reduce the mutual couplings of large antenna arrays [31]. However, the metasurface is based on resonant structures that cannot be extended for wideband applications. In addition, it is not feasible to be applied to dual-polarized arrays due to the asymmetric geometry along vertical and horizontal directions simultaneously. 

This paper proposes a non-resonant metasurface (NRMS) and places it above a 4 × 4 dual-polarized and large-scale antenna array for isolation enhancements. The NRMS can be equivalent to a medium with a negative permeability and a positive permittivity to suppress the coupling along the tangential direction of arrays. The isolation can be improved from 15 dB to 24 dB within 4.36–4.94 GHz after loading the proposed NRMS. Unlike the previous metasurface decoupling methods, the equivalent permeability and permittivity of the NRMS are extracted from the non-resonant working band. Thus, the NRMS element has an ultra-wide operating band with negative permeability for decoupling. The extracted parameters of the proposed NRMS are almost unchanged with different incident angles. Unlike the traditional metasurface [29,30,31], the proposed NRMS units in this paper are symmetric in two dimensions besides their wideband and wide-angle decoupling performance. Therefore, the proposed decoupling method can be utilized for large planar dual-polarized phased arrays with complex electromagnetic coupling paths. Due to its low profile, low complexity, and wide bandwidth, the proposed NRMS is a good candidate for reducing mutual couplings of dual-polarized and large-scale antenna arrays.

This paper is organized as follows. In Section 2, the operating mechanism of the NRMS is explained. Then, a procedure of a dual-element dual-polarized antenna array with NRMS is elaborated on to verify the effectivity of the proposed decoupling method. To verify the feasibility of the proposed decoupling method for large-scale antenna arrays, a 4 × 4 dual-polarized antenna array with NRMS is designed, and its corresponding performance, parametric study, and the comparison between our work and the techniques employed in the latest literature are presented in Section 3 as well. Section 4 provides the conclusions.

## 2. Non-Resonant Metasurface for Decoupling

The free-space wave coupling mainly causes the mutual coupling between massive antenna array elements with a half-wavelength distance. Therefore, the metasurface employed above the array is mainly used to reduce the free-space coupling path. This section will investigate the scheme of the proposed isolation improvement method in detail. The decoupling principle will be analyzed based on a massive MIMO antenna array sketch with NRMS. The NRMS unit cell is studied under TE and TM modes when the incidece waves propagate in various directions to analyze the effects on the extracted permittivity and permeability to investigate the design procedure of NRMS. Then, the decoupling principle is studied in detail with an example of dual-element dual-polarized antenna array.

### 2.1. Decoupling Scheme of the NRMS

The sketch of the isolation enhancement principle with NRMS is shown in Figure 1a. The space waves radiated from the element P3 can be broadly represented with a1 and a2, where a1 and a2 are responsible for the mutual coupling between adjacent and non-adjacent antenna elements, respectively. When the NRMS is placed above the array with a distance of h, the NRMS can be equivalent to a negative permeability and positive permittivity medium along the tangential (or x-axis) direction of the NRMS when the unit cell of the NRMS is properly designed, where the propagation constant is purely imaginary. As a result, the propagation of the a1 and a2 at the tangential direction of the array will be prohibited. Unlike the resonant-based MS that only works in a narrow band, this paper proposes a non-resonant and symmetric MS for wideband and dual-polarized large-scale antenna arrays. 

The geometry of the proposed NRMS unit cell is shown in Figure 1b, developed from the periodic cross-shaped ring, and an air cavity is engraved on the center of the NRMS. The metal strips with a 0.5 mm width are printed on the RO 4350B substrate with a permittivity of 3.66, a loss tangent of 0.002, and a thickness of 1.524 mm. 

### 2.2. Study for the NRMS Unit Cell

Figure 2a shows the simulation model of the unit cell of the NRMS when the incidence waves impinge on it normally, where the E-field of the incidence waves in the tangential direction of the NRMS and the H-field of the incident waves in the vertical direction of the propagation direction of the incident waves. Wave ports are embedded on the top and bottom surfaces of the unit cell without any air gaps. The simulated S−parameters under TE and TM modes at different incident angles of θ offset the normal direction are provided in Figure 2b. The S_11_ is less than −2.5 dB for TE mode, while it is lower than −6 dB for TM mode within 60° from 3 to 5 GHz. The extracted permittivity and permeability under TE and TM modes with different θ are given in Figure 2c,d, respectively. Both the extracted permittivity and permeability of the unit cell under TE and TM incidence waves at different angles of θ are positive, which means that the space waves along the normal direction of the NRMS can propagate through the unit cell freely.

Figure 3a illustrates the simulation model of the unit cell when the incidence waves propagate along the tangential direction of the unit cell, where the H-field of the incidence waves are in the vertical direction of the unit cell, and the E-field is perpendicular to the direction of the incidence waves. The wave ports are also embedded on the left and right sides of the unit cell. Figure 3b shows the simulated S-parameters under TM mode incidence wave at different incident angles of φ, where it shows the S_21_ is less than −9 dB. The extracted permittivity and permeability under TM incidence wave at different incidence angles can be found in Figure 3c,d. The NRMS unit cell exhibits a negative permeability from 3.0 to 5.1 GHz, while the extracted permittivity is positive within the same band. The S-parameters and the corresponding extracted equivalent parameters demonstrate that the space waves coupling under TM incidence wave cannot propagate along the tangential direction of the unit cell. It is also found that the extracted permittivity and permeability of the NRMS under TE mode are all positive, indicating that the NRMS cannot suppress the mutual couplings generated by TE modes. This specifies a design idea that an anisotropic NRMS can be properly configured to offer negative permeability and positive permittivity for both TE and TM modes to further reduce the mutual couplings among antenna elements, which is our future work. Owing to the features of wideband and geometry symmetric along the orthogonal directions, the proposed NRMS unit cell is promising to reduce the mutual couplings of wideband and dual-polarized large-scale antenna arrays.

### 2.3. Decoupling of A Two-Element Antenna Array with the NRMS

In this section, an example of a dual-element antenna array with a NRMS is given to verify the decoupling performance of the proposed NRMS. The design procedure of the isolation improvement for a dual-element dual-polarized antenna array with NRMS and the configuration of the reference array in the design procedure is given in Figure 4a,b, respectively. This array implements stacked microstrip antenna elements to obtain a broad operating bandwidth. The square metal patch is printed on the top surface of the RO 4350B substrate with a permittivity of 3.66 and a loss tangent of 0.002. The center-to-center distance between the array elements is 16.5 mm (0.5λ at the center frequency). Four orthogonal slots are etched on the bottom square patch to reduce the cross-polarized mutual coupling between two ports of the antenna array element itself. A PP (polypropylene) board with a permittivity of 2.2 is placed above the lower layer substrate to support the upper layer patch antenna. Then two cavities on the corresponding position of the antenna array elements are engraved to provide a space for the PCB board solder. Port1 and Port2 work in x-polarization, while Port3 and Port4 work in y-polarization. Here, the PP board does not have any other impact on the antenna performance besides supporting the two-layer substrates. The reference antenna array shown in Figure 4a is labeled as Array 1. In the next step, Array 2 is shown, where the NRMS is employed above Array 1. Here, the period of T and the size of x1 of the metal ring is 8 mm and 2.75 mm, respectively. The thickness of the substrate is 1.524 mm. The specific geometry and dimensions of the reference array in the design procedure are depicted in Figure 4b, including the overall structure of the reference array from the front side view and the structure of each layer from the top side view.

The S parameters of Array 1 and Array 2 are given in Figure 5. Port 1 and Port 2 work in y- and x-polarization, respectively. All S_13_ and S_42_ of the arrays with the proposed NRMS show low mutual coupling between the ports with the same polarization. Furthermore, Array 2 can provide a wider decoupling bandwidth. S_14_ and S_23_ of the array with NRMS present the mutual coupling between the ports with the cross-polarization. The simulated S-parameters also verify the theoretical analysis for the decoupling with the NRMS mentioned in the previous section. The optimized dimensions of the NRMS and the high *h* are listed in Table 1.

From the decoupling study of the two-element antenna array with the proposed NRMS, it can be concluded that the proposed NRMS can be equivalent to a negative permeability and positive medium along the tangential direction of the antenna arrays. Therefore, the NRMS will suppress the propagation of the free-space coupling component along the tangential direction. The best decoupling level can be achieved by carefully designing the sizes of the NRMS element and the height of the NRMS above the antenna array. 

## 3. Example of 4 × 4 Array with NRMS

### 3.1. Antenna Configuration

Sometimes, the decoupling method that is effective to the two-element antenna array does not necessarily work for large-scale antenna arrays (e.g., 4 × 4 antenna array, even larger), where much more complicated coupling paths exist in large-scale antenna array. As a result, the proposed NRMS is also utilized to check its feasibility to improve the isolation of a wideband and large-scale antenna array. For brevity, a wideband and dual-polarized 4 × 4 antenna array is investigated here. 

The antenna element and element dimension used in the stacked micro-strip antenna array in Section 2 are also applied to the 4 × 4 arrays in this section. The proposed large-scale antenna array that consists of 16 elements with an inter-element distance d of 33 mm covers the bandwidth from 4.29 to 5.13 GHz. The micro coaxial cables are adopted to excite the antenna elements of the antenna array. Different from the dual-element antenna array in Figure 4, the mutual couplings of the 4 × 4 phased arrays exist between the adjacent and non-adjacent elements in both co-polarization and cross-polarization. The mutual coupling between the array elements in the diagonal direction cannot be neglected either. The proposed NRMS, loaded above the antenna arrays with a height h of 15 mm, is expected to simultaneously reduce the mutual coupling of all the paths in a wideband. A design procedure of the NRMS is shown in Figure 6. Here, the original antenna array is labeled as Array A. The original array with the proposed NRMS is marked as Array B, where some non-metalized holes are drilled in both substrates, and a foam board is made with a thickness of 15 mm to support the NRMS. The simulated model of the 4 × 4 antenna array with NRMS is depicted in Figure 6a, where the NRMS is placed above the original antenna array with a distance. The detailed structure of the original antenna array and the NRMS is depicted as well. The prototype is depicted in Figure 6b, where all foam boards and substrates are compressed into one piece and fixed by screws and bolts passing through the non-metalized holes.

### 3.2. Parametric Study

The height of the NRMS is an important parameter to determine the decoupling level of the array. Therefore, a parametric study for the height h is essential to obtain the optimal decoupling level for the 4 × 4 large-scale antenna array. Figure 7 gives the S-parameters of the antenna array when h varies from 9 to 21 mm with a step of 3 mm. Here, the couplings between element A6 and the neighboring and non-neighboring elements are selected. It shows that, when h is 15 mm, the lowest couplings between antenna elements can be obtained, and the mutual couplings of the antenna array are lower than −24 dB. As h is set as other values, the mutual couplings of the antenna array are higher than 24 dB.

The size x1 of the metasurface cell (see Figure 1b) also plays an essential role in determining the decoupling level because the cross-shaped pattern of the NRMS unit cell fully determines the S-parameters and the corresponding extracted permittivity and permeability. Theoretically, the x1 and x3 determine bandwidth that we can extract negative permeability and positive permittivity. Thus, the parametric study for the size x1 and x3 for the decoupling level of the 4 × 4 large-scale antenna array is provided. Figure 8 gives the S-parameters when size x1 varies from 1.25 to 2.75 mm with a step of 0.5 mm. It depicts that, for the increment of x1, the mutual coupling of the 4 × 4 large-scale antenna array decreases. When x1 is 2.75 mm, the mutual coupling of the 4 × 4 large-scale antenna array reaches the lowest level, where the mutual coupling is lower than 24 dB. When x1 increases further, the more reflected waves from the proposed NRMS will be generated, which might enhance the space wave coupling of the array. Besides, the impedance between the proposed NRMS and antenna array will be deteriorated as well.

Figure 9 shows the S-parameters when size x3 varies from 0.6 to 6.6 mm with a step of 2 mm. It clearly demonstrates that, with the increment of x3, the mutual coupling of the 4 × 4 large-scale antenna array reduces slightly. When the air cavity size x3 of the NRMS unit cell is 6.6 mm, the lowest mutual coupling level of the antenna array can be achieved. When x3 increases further, the air cavity will destroy the structure of cross-shaped NRMS units. Thus, the S-parameters of the 4 × 4 large-scale antenna array with larger air cavity size x3 (larger than 6.6 mm) are not given. Finally, the optimized x1 and x3 is 2.75 mm and 6.6 mm, respectively. Comparing the parametric study of x1 and x3 on the decoupling level of the 4 × 4 large-scale antenna array, it could conclude that the size of the cross-shaped structure plays a determining role, and the size of the air cavity plays a fine-tuning function.

### 3.3. Antenna Array Performance

The 4 × 4 arrays in Figure 6 have the mutual coupling between adjacent and non-adjacent array elements in the x, y, and diagonal directions. Moreover, the arrays are symmetric along the x, y, and diagonal directions. Due to the highly geometry symmetric of the NRMS and dual-polarized antenna array, the mutual coupling between the array element A6 (see Figure 6a) and other elements can represent all coupling types. Here, the S-parameters of A6 are selected and shown. S_11,12_ represents the mutual coupling level between port11 and port12 of the array element A6. It should be noticed that S_2,12_ refers to the mutual coupling between neighboring array elements in the diagonal direction. S_4,12_ and S_10,12_ represent the mutual couplings between adjacent elements in the y-direction and x-direction, respectively. In addition, S_16,12_ and S_28,12_ refer to the mutual coupling between non-neighboring array elements. The S-parameters of port11, similar to that of port12, can also represent the mutual coupling of the proposed 4 × 4 large-scale antenna array. 

Figure 10 gives the simulated and measured S-parameters of the 4 × 4 antenna array with and without the proposed NRMS. In Figure 10a, the reference array covers from 4.05 to 4.91 GHz with an isolation of 16.5 dB. Figure 10b shows that the array with the proposed NRMS covers 4.36 to 4.94 GHz with an isolation of 24 dB. The measured S-parameters of the antenna array with the proposed NRMS in Figure 10c align well with the simulated one in Figure 10b. Therefore, all simulated and measured S-parameters demonstrate that the isolations of the wideband and dual-polarized large-scale antenna array can be improved effectively to over 24 dB by employing the proposed NRMS within 4.36 to 4.94 GHz.

The simulated and measured radiation patterns of the array before and after loading the proposed NRMS at 4.5 GHz, 4.7 GHz, and 4.9 GHz are shown in Figure 11 to check the impacts of the proposed NRMS on the radiation performance of the antenna element. The measurements are implemented in the anechoic chamber to avoid electromagnetic interference from the environment. As seen in Figure 11, the radiation patterns of the antenna arrays with and without the proposed NRMS are almost unchanged except for some slight ripples. Moreover, the cross-shaped metal rings convert partial space wave energy from one polarization into orthogonal polarization, which causes the deterioration of the cross-polarization level of the antenna elements. Additionally, the radiation patterns are asymmetric due to the propagation blockage of the space waves radiated from the active antenna by its adjacent antennas.

Finally, the realized gains and total efficiencies of the antenna array with and without the proposed NRMS are presented in Figure 12. The simulated results illustrate that the proposed NRMS can improve the realized gain and total efficiency of the antenna array within a wide bandwidth, which are attributed to the suppressed propagation of the space waves coupling along the tangential direction. Moreover, the measured realized gain and total efficiency of the array with the proposed NRMS are lower than the simulated results caused by the measurement error. The losses in coaxial cables also cause the discrepancies between the simulation and the measurement of the antenna arrays. Here, the measured realized gain and total efficiency are more than 6 dBi and 74% from 4.36 to 4.94 GHz, respectively.

### 3.4. Antenna Performance Comparison

The performance of the proposed 4 × 4 large-scale antenna array with the proposed NRMS is compared with the literature reported recently, as listed in Table 2. The decoupling proposed in [24] is an effective method, but it has a fatal defect of a narrow bandwidth caused by the resonant–response property of the decoupling network. Typically, it needs to be deployed on the backside of the antenna array, where the antenna array with the decoupling network has multiple substrates, which means they are bulky and complicated. Moreover, the decoupling network usually lowers gain and total efficiency of antenna array. The ADS in [26] is a novel decoupling concept, but it has a complicated design process for the pattern of the primary reflectors and secondary reflectors. Additionally, it needs a relatively larger space to accommodate the reflectors to reflect enough space waves so that the unwanted coupling waves can be largely eliminated, which is not conducive to antenna miniaturization. Though the gain and total efficiency of the array in [28] are higher than that of our work, it has a larger inter-element distance and a bulky structure. The metasurfaces proposed in [29,31] have a relatively small inter-element distance, but they can only be used in single-polarized arrays because of the magnetic resonant response and the asymmetry along the cross direction. Moreover, the bandwidth, gain, and total efficiency of the arrays in [29,31] are worse than our work. Additionally, the worst isolation of the array in [31] is higher than our work, but it cannot be expanded to a massive MIMO array application owing to the limitation of asymmetrical structure along the orthogonal directions. Compared with the structure in the current literature, on the premise of guaranteeing a wider working bandwidth, higher worst isolation, gain, and efficiency, our work has a simple design and installation process of the decoupling scheme. 

## 4. Conclusions

A novel decoupling concept of NRMS for wideband and dual-polarized large-scale antenna arrays is proposed. The decoupling mechanism of the NRMS has been analyzed. To justify the feasibility of the proposed NRMS for decoupling of large-scale antenna arrays, a 4 × 4 antenna array loading with the proposed NRMS has been simulated, fabricated, and measured. The simulated and measured results demonstrate that the isolations of the antenna array can be enhanced from 16.5 dB to over 24 dB within the band from 4.36 to 4.94 GHz, almost without introducing any other adverse effect on the performance of the overall antenna array. The comparison between the proposed NRMS and other techniques reported in the latest literature shows a great potential of the proposed NRMS to be applied to massive MIMO antenna arrays with low mutual couplings in the sub-6 GHz band.

## Figures and Tables

**Figure 1 sensors-23-00152-f001:**
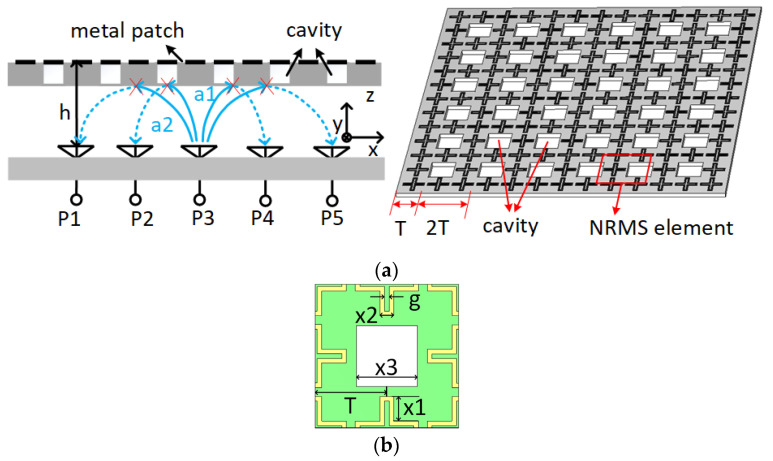
(**a**) the decoupling principle of the NRMS for the large-scale antenna array, (**b**) the detailed structure and dimensions of the unit cell to implement the NRMS.

**Figure 2 sensors-23-00152-f002:**
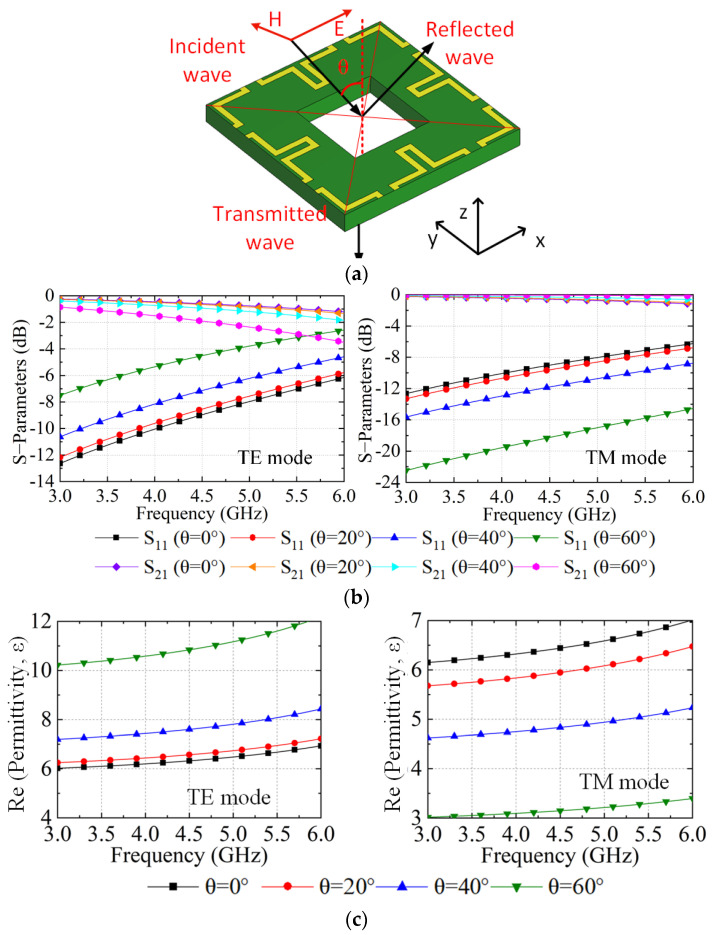
NRMS unit cell with incident angle along the normal direction of the surface: (**a**) the simulation model, (**b**) the S-parameters under TE and TM incidence waves at different incident angles, (**c**) the real part of the permittivity under TE and TM incidence waves at different incident angles, and (**d**) the real part of the permeability under TE and TM incidence waves at different incident angles.

**Figure 3 sensors-23-00152-f003:**
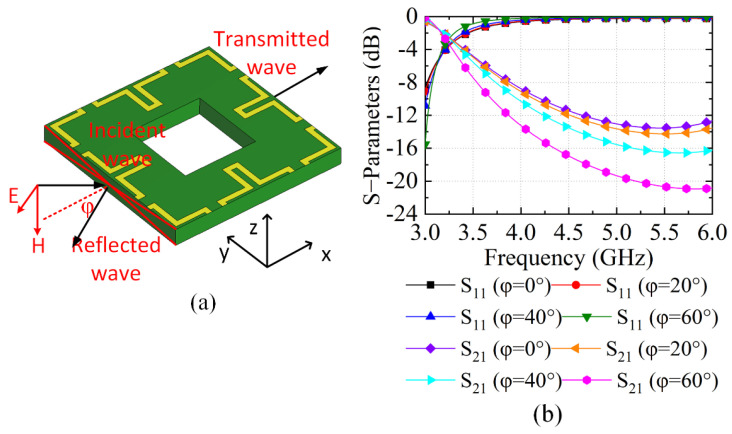
NRMS unit cell with incident angle along the tangent direction of the surface: (**a**) the simulation model, (**b**) the S-parameters, (**c**) the real part of the permittivity in different incident angles, and (**d**) the real part of the permeability in different incident angles.

**Figure 4 sensors-23-00152-f004:**
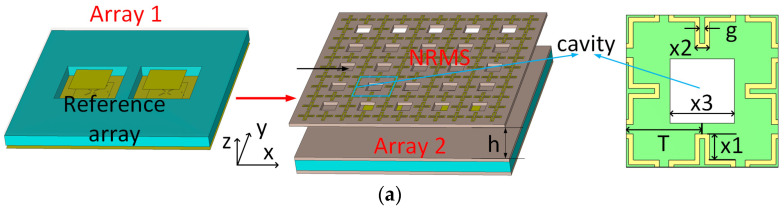
(**a**) Design procedure of the isolation improvement for dual-element dual-polarized antenna array with NRMS. Array 1 is the reference antenna array. In the next step, Array 2 is depicted, where a metasurface consisting of periodic cross metal rings with a period T is employed above Array 1, where the air cavities with a period 2 × T are engraved on the metasurface. (**b**) The configuration of the reference array in the design procedure. (Unit: mm).

**Figure 5 sensors-23-00152-f005:**
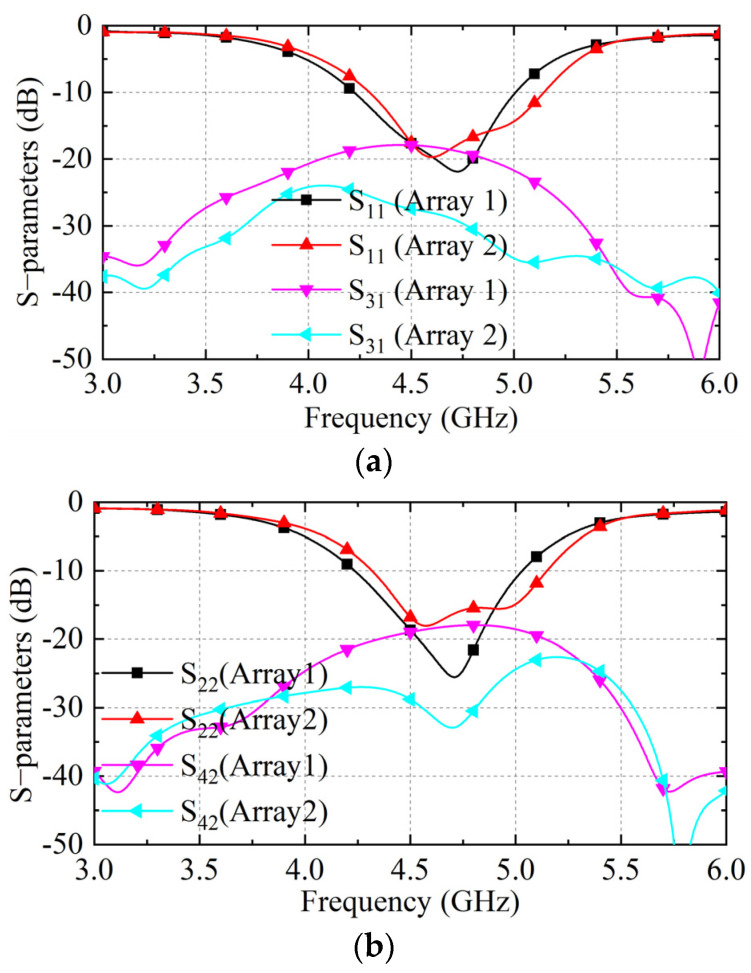
The S parameters of array1 and the one after adding MS and NRMS for array2 and array3, (**a**) the S_11_ and S_31_, (**b**) S_22_ and S_42_, and (**c**) S_14_ and S_23_.

**Figure 6 sensors-23-00152-f006:**
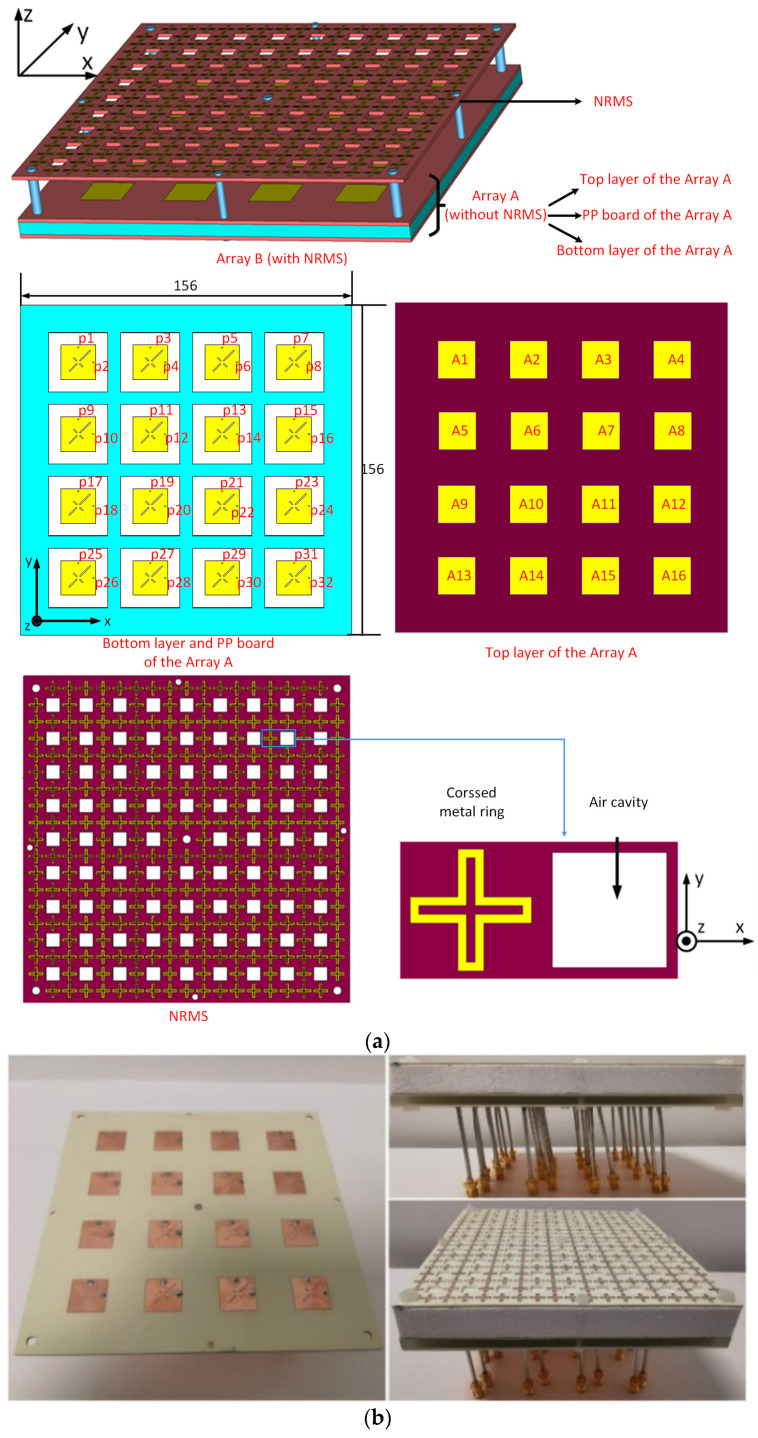
(**a**) the configuration of the original 4 × 4 phased array with NRMS (Array B), (Unit: mm). (**b**) the fabricated prototype of the Array B with feeding cables.

**Figure 7 sensors-23-00152-f007:**
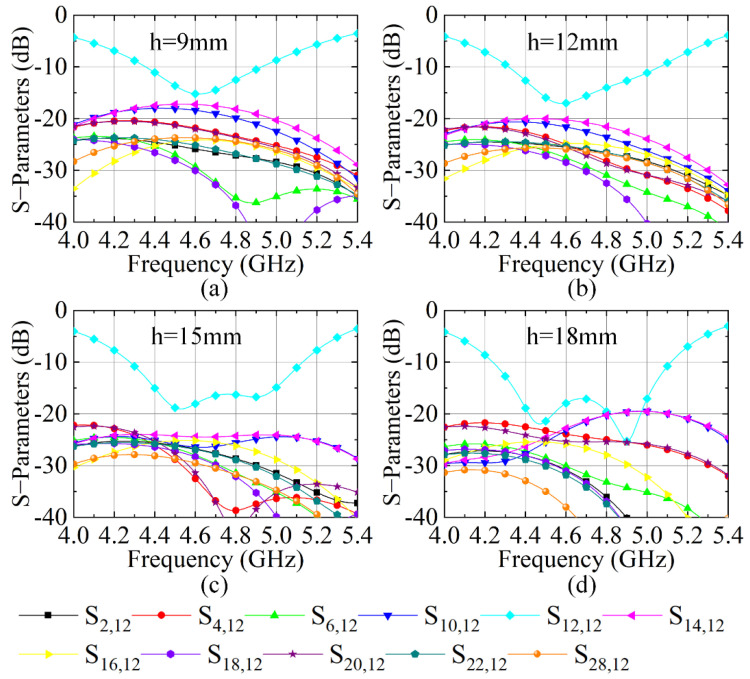
S-parameter of the antenna array loading with the proposed NRMS with different h, (**a**) h = 9 mm, (**b**) h = 12 mm, (**c**) h = 15 mm, and (**d**) h = 18 mm.

**Figure 8 sensors-23-00152-f008:**
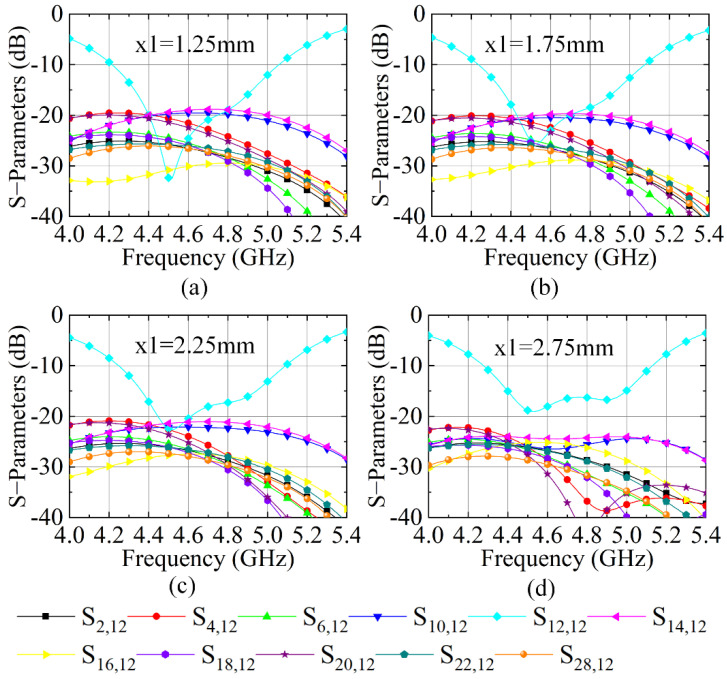
S-parameter of the antenna array loading with the proposed NRMS with different x1, (**a**) x1 = 1.25 mm, (**b**) x1 = 1.75 mm, (**c**) x1 = 2.25 mm, (**d**) x1 = 2.75 mm.

**Figure 9 sensors-23-00152-f009:**
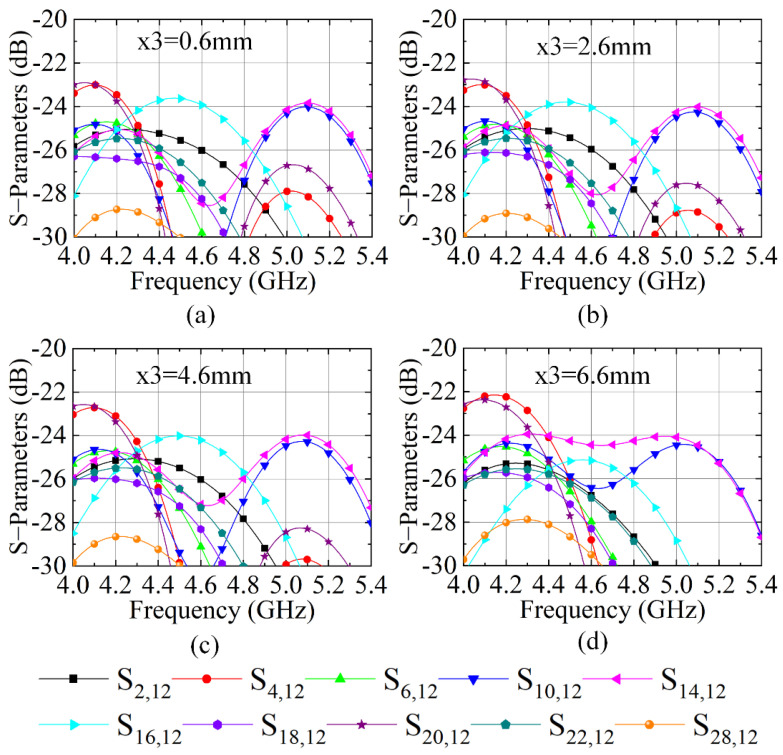
S-parameter of the antenna array loading with the proposed NRMS with different x3, (**a**) x3 = 0.6 mm, (**b**) x3 = 2.6 mm, (**c**) x3 = 4.6 mm, (**d**) x3 = 6.6 mm.

**Figure 10 sensors-23-00152-f010:**
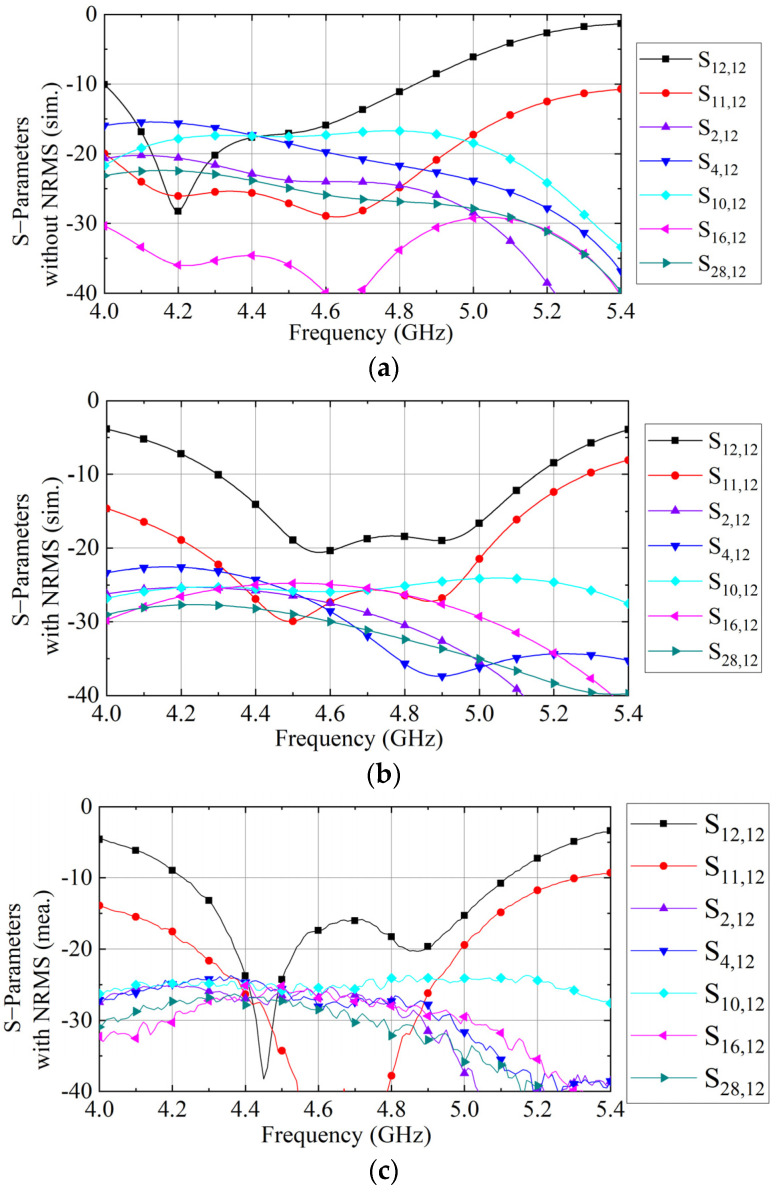
(**a**) S-parameters of arrays with and without NRMS, (**a**) the simulated S-parameters without NRMS, (**b**) the simulated S-parameters with NRMS, and (**c**) the measured S-parameters with NRMS.

**Figure 11 sensors-23-00152-f011:**
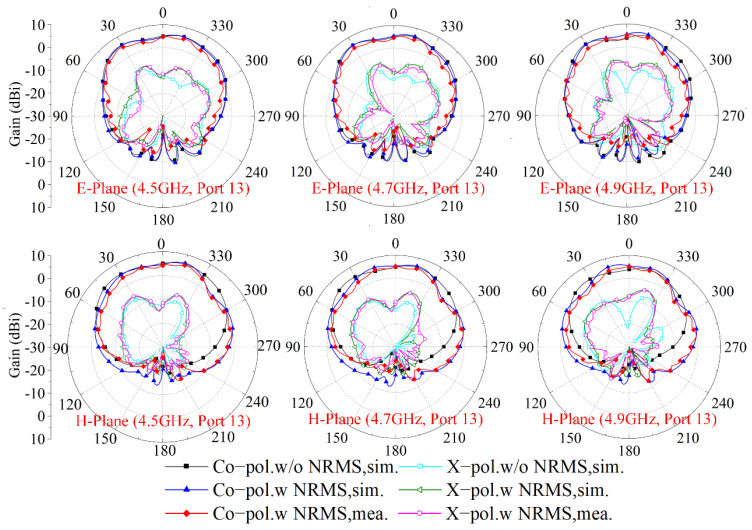
The radiation patterns of the array with and without NRMS at different frequencies.

**Figure 12 sensors-23-00152-f012:**
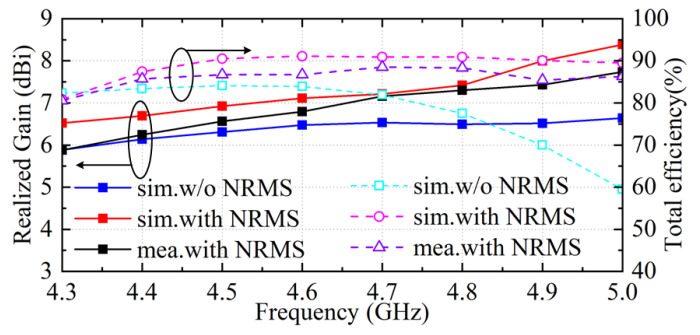
The total efficiency and the realized gain of the proposed antenna array.

**Table 1 sensors-23-00152-t001:** Parameters of the NRMS.

Parameters	x1	x2	x3	g	T	h
Value (mm)	2.75	1.5	6.6	0.5	8	15

**Table 2 sensors-23-00152-t002:** Performance comparison of the proposed antenna arrays with other state-of-the-art similar works.

Ref.	Decoupling Method	Pol. and Scale	Freq. (GHz)	Worst Iso. (dB)	Gain (dBi)	Total Efficiency	Antenna Distance and Height (λ_0_)	Feasib. for Massive MIMO Arrays	Compl.
[24]	Decoupling network	Dual-pol. 4 × 4	4.85–4.95 (2.0%)	25	5.3	>70%	0.50λ_0_, 0.274λ_0_	Yes	High
[26]	ADS	Dual-pol. 4 × 4	3.3–3.8 (14.1%)	25	6.0	--	0.64λ_0_, 0.4λ_0_	Yes	High
[28]	DG	Dual-pol. 4 × 4	4.9–5.2 (6.1%)	25	7.3	>90%	0.62λ_0_, 0.25λ_0_	Yes	High
[29]	Metasurface	Single-pol. 2 × 1	5.49–6.0 (8.64%)	27	5.0	>63%	0.259λ_0_, 0.180λ_0_	No	Low
[31]	Metasurface	Single-pol. 4 × 4	5.67–5.97 (5.17%)	19	5.0	>70%	0.43λ_0_, 0.19λ_0_	Yes	Low
**This work**	**NRMS**	**Dual-pol.** **4 × 4**	**4.36–4.94 (12.5%)**	**24**	**6.0**	**>74%**	**0.5λ_0_, 0.38λ_0_**	**Yes**	**Low**

Ref.: reference; Pol.: polarized; Freq.: frequency; Iso.: isolation; Feasib.: feasibility; Compl.: complexity.

## Data Availability

Not Applicable.

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
