# Peer review of "Decoupling of Dual-Polarized Antenna Arrays Using Non-Resonant Metasurface"

_sensors, 2022, doi:10.3390/s23010152_

Round 1
Reviewer 1 Report
The paper present a decoupling method for dual-polarized antenna arrays by introducing the non-resonant metasurface. The simulated and measured results demonstrated the proposed method is effective to suppress the mutual coupling between antenna elements. The comment are listed below:
Q1. The directions of the E-field, H-field and the wave propagation in Fig.2 (a) and Fig. 3(a) have some errors. Theoratically, they should satisfy the right-hand screw rule.
Q2. The caption for Fig.5 is not clear. It is difficult to read which one is without NRMS.
Q3. In Fig.3, only the dielectric paremeters of the TM are given, why did not give the TE mode.
Author Response
Dear Reviewer,
Thanks a lot for your insight suggestions. All comments have been carefully addressed.

Reviewer 2 Report
I recommend accepting the paper as it is.
Author Response
Dear Reviewer,
thanks a lot for your positive suggestions.

Reviewer 3 Report
A non-resonant metasurface (NRMS) concept is reported in this paper to improve the isolation of dual-polarized and wideband large-scale antenna arrays. Overall, the paper is well written and the result is interesting, but some issues in the manuscript should be clarified for better understanding.
In particular:
1) The logic in Section 3 could be improved. I suggest that the parameter study could move to the beginning of Section 3.
2) In the caption of Fig. 4, the ‘array3’ is mentioned in the step B. However, array3 does not appear in Figure 4.
3) Still in Figure 4, please provide a layer view of the reference antenna. In this way, the reader could understand the description of the antenna better.
4) The ‘(a)’ at the beginning of the Fig. 6 caption should be removed.
5) For improved isolation performance, it would be better to provide an S21 difference between the array with and without NRMS.
6) Maybe more mutual coupling conditions should be considered, i.e., A1 and A13, A1 and A14, A1 and A15, and A1 and A16.
Author Response
Dear Reviewer,
Thanks a lot for your kind suggestions. All comments have been carefully addressed.
